# Effects of the Grapevine Biochar on the Properties of PLA Composites

**DOI:** 10.3390/ma16020816

**Published:** 2023-01-13

**Authors:** Chien-Chung Huang, Chun-Wei Chang, Kousar Jahan, Tzong-Ming Wu, Yeng-Fong Shih

**Affiliations:** 1Department of Applied Chemistry, Chaoyang University of Technology, Taichung 41349, Taiwan; 2Department of Aeronautical Engineering, Chaoyang University of Technology, Taichung 41349, Taiwan; 3Department of Materials Science and Engineering, National Chung Hsing University, Taichung 402, Taiwan

**Keywords:** PLA, grapevine char, degradation, agricultural waste

## Abstract

This study found that biochar made from grapevines (GVC), an agricultural waste product, can be used as a nucleating agent to promote the crystallization of polylactic acid (PLA). Differential scanning calorimetry (DSC) analysis of GVC/PLA composites showed that different particle sizes (200 and 100 mesh size) and amounts (1 wt%, 10 wt%) of biochar affect the re-crystallization of PLA, with 200 mesh GVC in the amount of 10 wt% being the most significant. In addition, it was found that there were two peaks related to imperfect and perfect crystals in the T_m_ part for GVC/PLA composites. TGA analysis showed that adding GVC tends to lower the maximum decomposition temperature of PLA, revealing that GVC may accelerate the degradation reaction of PLA. This research also studied the effects of GVC in various particle sizes and amounts on the mechanical properties and degradation of PLA. The results revealed that the tensile and impact strengths of GVC/PLA composite could reach 79.79 MPa and 22.67 J/m, respectively, and the increments were 41.4% and 32.1%, greater than those of pristine PLA. Moreover, the molecular weight of PLA decreased as the amount of GVC increased. Therefore, GVC particles can be used as reinforcing fillers for PLA to improve its mechanical properties and adjust its molecular weight. These agricultural-waste-reinforced biocomposites can reduce both greenhouse gas (GHG) emissions and the cost of biodegradable polymers and achieve the goals of a circular economy.

## 1. Introduction

Traditional plastics, including thermoplastics and thermosets, are used widely in all aspects of life due to their affordability, light weight, strength, and durability. However, more than 300 metric tons of conventional plastic waste are accumulated annually due to its slow degradation rate. To reduce the use of fossil-derived plastics and obtain sustainable and environmentally friendly substitutes for petrochemicals, many professionals directed their attention to the development of biodegradable polymers [1,2,3]. Biodegradable polymers, such as polylactic acid (PLA), polybutylene succinate (PBS), polycaprolactone (PCL), and polybutylene adipate terephthalate (PBAT), have been successfully commercialized [4,5] as packaging materials in the past. However, due to the demand for net zero carbon emissions, biodegradable polymers are also being explored in other industries, such as automobiles, electronics, and biomedical engineering [6,7,8]. Because of its extensive benefits, including low energy consumption, environmental friendliness, high transparency, biocompatibility, biodegradability, and reliable mechanical properties [9,10,11], PLA has become one of the most commercialized biobased synthetic polymers and accounts for more than 25% of global biodegradable polymer production. 

Agricultural waste is an important source of biofuels that can offer an adequate substitute for fossil fuels. It can not only be used to produce renewable energy, such as bioalcohols, but can also be utilized to produce various biochemicals, materials, and fertilizers. Therefore, agricultural waste can reduce resource consumption to a certain extent and meet the goal of a circular economy. No Agricultural Waste (NoAW), a research and development project funded by the European Union’s Horizon 2020, supports the “zero waste economy” in response to current pressing environmental problems. As an international organization with a vast reach, NoAW aims to develop innovative technologies that fully utilize agricultural waste. NoAW plans to assist policymakers and business owners in transforming today’s agricultural waste into tomorrow’s value-added products. NoAW also investigates how agricultural wastes can be converted into various eco-friendly products while minimizing the use of natural resources and reducing environmental pollution [12]. In its 2030 agenda for Sustainable Development, the United Nations listed “sustainable agriculture” as one of its major goals. Circular agriculture is the efficient use of agricultural crops in combination with material recycling technologies to reduce waste and greenhouse gas emissions.

Global wine consumption amounted to 236 million hectolitres in 2021, with grapes being the most preferred raw materials for wine production. This contributed to a huge amount of grapevine waste (>42 million tons) generated annually worldwide [13]. Every year, Spain generates more than 2 million tons of grapevine waste. These wastes include grape leaves and stems, grape skins and pomace, lees, and wastewater coming from wine-making industries. Waste utilization includes using lees as animal feed and recycling and reusing wastewater. However, grape leaves and stems are ineffectively used and are often discarded as waste [14,15,16]. In 2020, Taiwan’s annual agricultural waste reached as high as 5.34 million tons, posing serious threats to animals, humans, and the environment if left unmanaged [17]. Currently, most of the pruned grapevines from Taiwan’s wine-making industries are discarded. These grapes wastes are rich in lignocellulosic material and can be used to make biochar or grapevine char (GVC). Due to its adsorption ability, agricultural-waste-derived biochar can be a promising tool for the remediation of heavy metals, pesticides, herbicides, and pharmaceuticals, and it can also reduce nitrous oxide (N_2_O) and methane (CH_4_) flux from soils. Moreover, the stability of biochar in soils falls mostly in the centennial to millennial time scales, suggesting that it can serve as a durable carbon sink [18,19]. In addition, biochar can be a green filler used to improve a polymer’s thermal conductivity, hardness, and heat resistance, allowing polymer products to have reduced carbon emissions [20,21,22]. 

There have been few research studies on biochar-reinforced biodegradable polymers [23,24,25,26]. Arrigo et al. utilized the biochar derived from spent ground coffee to reinforce PLA and found that PLA degrades at high temperatures during the melt mixing process [24]. As is well known, the degradation rate of PLA is not fast enough at room temperature and in a natural environment. However, PLA degradation is much faster in controlled conditions at the industrial scale than in home composting conditions. Moreover, PLA-based containers and cups were found to be fully biodegradable, taking 1~3 months and six months to compost at an industrial facility and in home conditions, respectively [27,28]. As a result, speeding up the degradation process of PLA attracted colossal attention from researchers, who conducted several studies in this area. Aguirre et al. demonstrated the use of a bioaugmentation method with a PLA-degrading microbial strain *Geobacillus thermoleovorans* [28] to accelerate the degradation of PLA. Other studies evaluated the effects of biomass fillers, such as agricultural fibers and algae, on the biodegradability of PLA [29,30]. Kalita et al. conducted aerobic biodegradation tests of PLA/algae composite following ASTM International standard D5338-15. They found that the 5 wt% of deoiled algae biomass significantly improved the biodegradation behavior of the PLA samples. This could be attributed to the nitrogen-rich algae biomass that hastened the degradation rate of PLA by promoting microbial growth [30].

This study used biochar derived from GVC to reinforce the thermal and mechanical properties of PLA and investigate its effects on adjusting the PLA decomposition. These GVC-reinforced PLA composites are expected to be used to produce agricultural equipment and tools, such as clips, labels, and supports for growing grapes or other plants to promote recycling and reuse of agricultural resources. Moreover, these biodegradable facilities can be recycled or composted with plant residues. The application of agro-wastes as reinforcement for preparing biocomposites yields benefits such as biodegradability, reduced GHG emission reduction, reduced cost of biocomposites, and achievement of economic objectives [31,32].

## 2. Materials and Methods

### 2.1. Materials

NatureWorks, LLC provided the polylactic acid (PLA, PLA2003D) with a density of 1.24 g/cm^3^, melt flow index of 6.0 g/10 min at 210 °C, and a load of 2.16 kg. The grapevine (Figure 1a) was obtained from Dacun Township, Changhua County, Taiwan. HPLC grade Tetrahydrofuran (THF) was procured from Sigma.

### 2.2. Sample Preparation

#### 2.2.1. Preparation of Grapevine Char (GVC)

First, the grapevine was dried and cut into small pieces no larger than 5 cm. The small pieces of grapevine were then pyrolyzed in a muffle furnace at 500 °C in a nitrogen atmosphere for 2 h to produce the GVC. The pyrolyzed products, shown in Figure 1b, were crushed and ground before sieving through 120 and 200 mesh sieves to obtain the different sizes of GVC powder (120GVC and 200GVC), respectively. The GVC obtained had a moisture content of 5.36%, a pH level of 9.5, an ash content of 4.8% (mainly potassium and silicon), a volatility of 3.4%, a total carbon content of 92%, and a surface area of 105 m^2^/g.

#### 2.2.2. Preparation of GVC/PLA Composites

First, the PLA and GVC powder were dried at 80 °C in a vacuum oven for 12 h. Both PLA and GVC powders were then pre-melted at 175 °C for five minutes in a counter-rotating internal mixer (Brabender PL2000, Duisburg, Germany) with a rotation speed of 50 rpm. The mixing chamber (volume: 200 cm^3^) was filled with 200 g total mass, corresponding to an 80% filling ratio. After being well blended, the composite was granulated, and the GVC/PLA composite test specimens were prepared in a compression molding at 175 °C, with a pressure of 1.47 MPa for 5 min, 2.94 MPa for 3 min and 5.88 MPa for 1 min, in sequence. The PLA composites mixed with 1 and 10 wt% of 120 mesh GVC were labeled as PLA/120GVC-1 and PLA/120GVC-10, while the PLA composites mixed with 1 and 10 wt% of 200 mesh GVC were labeled as PLA/200GVC-1 and PLA/200GVC-10, respectively.

### 2.3. Characterization and Measurement

#### 2.3.1. Thermal Analysis

TA instrument DSC Q20 Differential Scanning Calorimeter was used to examine the thermal transition behavior. Samples of 5–10 mg were heated from room temperature to 190 °C at a heating rate of 10 °C/min and a nitrogen flow rate of 50 cm^3^/min. Calibration was performed using the iridium standard. Thermal decomposition behavior was determined via thermogravimetric analysis using TGA Q50, TA Instruments, Inc., Delaware, DE, USA. All samples were preheated at 70 °C in a hot air oven to remove moisture and then scanned from 50 to 600 °C at a heating rate of 10 °C/min under nitrogen flow. 

#### 2.3.2. Mechanical Properties Analysis

To study the mechanical performance of the PLA composites, tensile tests were performed in accordance with the ASTM D638 test method and using an Instron universal tester (HT-9102, Hung Ta Instrument Co., Ltd., Taichung, Taiwan). All the prepared samples were conditioned before testing at 65% relative humidity and 23 °C until equilibrium moisture content (EMC) was obtained. A 186 mm × 19 mm × 3 mm dumbbell-shaped test specimen was then prepared and tested at room temperature, with a load cell of 20 kN and an extension rate of 5 mm/min. Meanwhile, the impact strength of the composite material was studied using an impact resistance testing machine (GT-70045-MDL, Gotech Testing Machines Co., Taichung, Taiwan), with an initial potential energy of 2.7 J at a specified drop height of 610 ± 2 mm, following ASTM D256. A GT-HV 2000 analyzer was used to measure the heat deflection temperature (HDT) of the composites in accordance with the ASTM D648 test method. All the tests were performed for at least five specimens per sample, and the average values were calculated.

#### 2.3.3. Molecular Weight Analysis

Gel Permeation Chromatography (GPC, Waters 515 HPLC pump and Waters 717 Plus Autoinjector) was used to evaluate the variation in the molecular weight of PLA after compounding with GVC at 175 °C using a counter-rotating internal mixer. First, the required amount of test samples was dissolved in tetrahydrofuran (THF) at room temperature under ultrasonication for 10 minutes to obtain a final concentration of PLA composites of 0.1% (*w*/*v*). The solution was then centrifuged using ATO-HL-11KS (1200 rpm, 10 min) and was filtered with a 0.22 μm Teflon filter to remove the insoluble content and impurities before injecting the sample into the chromatographic columns. The mobile phase of GPC was THF with a flow rate of 1.0 mL/min at 40 °C. Calibration of the GPC was performed with polystyrene standards. The millennium analysis software was used to analyze the weight-average molecular weight (M_w,_ g/mol), the number-average molecular weight (M_n,_ g/mol), as well as the polydispersity index (PDI).

#### 2.3.4. Morphological Analysis of the Composites

The dimensions and surface shape of GVC were observed using an optical microscope (Carl Zeiss Axio Scope A1) with a digital camera. X-ray diffractometry (XRD) measurements were performed using a Rigaku D/MAX-2200PC X-ray diffractometer (Cu Kα radiation, wavelength: 0.154 nm) operated at 40 kV and 100 mA. Data were collected within the range of scattering angles (2θ) of 10–60°. The morphology of the GVC and composites were examined using SEM (HITACHI S-3000N, Hitachi Ltd., Tokyo, Japan).

## 3. Results and Discussion

### 3.1. Analysis of Biochar

Figure 2a–e shows the particle size distribution, SEM, TGA, and X-ray Diffraction (XRD) analysis of the GVC biochar. The particle sizes of 120GVC and 200GVC were 97~118 and 37~56 µm, respectively, as shown in the POM analysis in Figure 2a,b. The morphology of GVC (Figure 2c) exhibited a porous structure. TGA analysis (Figure 2d) showed that the char yield was high as 98%, indicating a low organic content. On the XRD patterns of GVC (Figure 2e), the graphitic peaks were found around the 2ϴ values of 25°(002) and 44°(10) [33]. However, the low intensity and inconspicuous peak revealed that the GVC biochar was an amorphous structure.

### 3.2. DSC Analysis

The DSC thermograms shown in Figure 3 illustrate the phase transition of the PLA and GVC/PLA composites during the heating process: the derived transition temperatures (T_g_s), cold crystallization temperature (T_cc_), and melting temperature (T_m_), as summarized in Table 1. As seen in Figure 3, the T_g_s of the GVC/PLA composites (58.03, 57.78, 53.42, and 53.42 °C) were lower than the control PLA (60.86 °C). Moreover, the T_g_ decreased as the amount of GVC increased. The decomposition of the polymer chain may lead to a lower transition temperature. Cold crystallization peaks ranging from 97.85 to 122.11 °C were found for all composites except for the pristine PLA. This suggests that when GVC is added to PLA, it acts as a nucleating agent, accelerating the crystallization of PLA upon heating. In addition, cold crystallization temperatures (110.83 and 97.85 °C) of 200 mesh GVC-containing composites (PLA/200GVC-1 and PLA/200GVC-10) were lower than those (122.11 and 108.28 °C) of 120 mesh GVC-containing ones (PLA/120GVC-1 and PLA/120GVC-10). Moreover, as the amount of GVC in the composite increased, the cold crystallization temperature decreased. This reveals that the nucleating effect of GVC on promoting the crystallization rate of PLA was more pronounced in the composite with a small particle size (200 mesh) and a higher amount (10 wt%) of GVC. Furthermore, two T_m_s (T_m1_ and T_m2_) were obtained in the thermograms of all the PLA/GVC composite except PLA/120GVC-1, with T_m1_ being lower than the T_m2_. PLA is known to have two different crystal structures, α and α’, with different melting temperatures. The formation of α and α’ crystal structures depends on their thermal history. The melting temperature of α’ crystals is lower than that of α phase [34]. Table 1 shows that the T_m1_s of the GVC/PLA composites (147.02, 135.10, and 140.12 °C) were lower than that of the pristine PLA (151.09 °C). This corresponds to the formation of α’ crystals and the decomposition of the polymer chain caused by the addition of GVC. Moreover, the T_m2_s (145.66 and 148.62 °C) of the PLA/200GVC-10 and PLA/120GVC-10 were much lower than T_m_ of the pristine PLA. Based on the T_g_ analyses, it was suggested that high loading of GVC could significantly decompose the polymer chain, resulting in a lower melting point. It should be noted that T_m1_ was not found for PLA/120GVC-1. Following the cold crystallization analysis, the nucleating effect was least obvious at lower content of 120 mesh GVC. Therefore, T_m1_ was not significant for PLA/120GVC-1. Furthermore, the crystallinity analysis revealed that PLA crystallinity increased with the GVC content (from 2.62% to 40.39, 22.09, 49.76, and 57.44%), indicating that GVC serves as an important nucleating agent.

### 3.3. TGA Analysis 

Figure 4 and Table 1 show the results of the investigation performed to determine the thermal stability of the PLA and GVC/PLA composites. GVC/PLA composites had lower thermal decomposition temperatures (T_d_) than pure PLA (except PLA/120GVC-1). This is consistent with the DSC analysis results, which suggest that adding GVC may cause the decomposition of PLA and lead to a lower T_d_. However, for PLA/120GVC-1, the acceleration of PLA decomposition was not so obvious with low content of GVC. This reveals that GVC can also act as an inorganic heat resistance filler and nucleating agent, resulting in only one obvious T_m_, so its T_d_ increased. Meanwhile, the significant reduction in T_d_ for PLA/200GVC-10 and PLA/120GVC-10 was caused by the relatively obvious acceleration of PLA decomposition at a high content of GVC. The molecular weight analysis confirms that the PLA chain was decomposed by the GVC catalyzation, resulting in decreased decomposition temperature. Moreover, the results showed that the char yield of the composites increased as the content of GVC increased. The char yields of the 120 mesh-containing composites were larger than those of the 200 mesh-containing ones. This is most likely due to the easier aggregation of the larger size filler in PLA, which has a smaller surface area. Hence, the heating was heterogeneously distributed along the PLA matrix [35]. Furthermore, DTG analysis (Figure 4b) revealed that PLA and all GVC/PLA composites have a single sharp decomposed peak. This suggests that PLA or all components of GVC/PLA composites decomposed simultaneously in a single reaction and that the addition of GVC had no significant effects on the decomposition behavior of PLA.

### 3.4. Mechanical and Thermal Analysis

Table 2 shows the results of the investigation aiming to study the mechanical and thermal properties of the PLA and GVC/PLA composites. The tensile strengths (65.98 and 79.79 MPa) of 1% GVC-containing composites (PLA/200GVC-1 and PLA/120GVC-1) were higher than those of pristine PLA (56.44 MPa). However, 10% GVC-containing composites (PLA/200GVC-10 and PLA/120GVC-10) exhibited a decline (41.72 and 29.30 MPa) compared to raw PLA. These findings indicate that the reinforcing effect of 120 mesh GVC on the tensile strength of PLA was better than that of the 200 mesh GVC at lower content (10%). Kamonwan et. al. demonstrated the effect of adding 0.25% different biocarbon on the properties of PLA, and the results showed that the tensile strength of the composites dropped from 56.63 MPa to 50.741~53.256 MPa when biocarbons were added [25]. In comparison with these results, it is clear that GVC has an excellent reinforcing effect. However, at high GVC content, the accelerating PLA degradation effect and uneven dispersion of GVC may result in a decrease in tensile strength. In addition, the elastic modulus of PLA was increased from 3.725 to 4.802~5.422 GPa with the addition of GVC. This revealed that the addition of GVC improved the stiffness of PLA. As shown in Figure 4 and Table 2, the impact strengths (20.00, 22.67, 15.33, and 17.11) of GVC/PLA composites were all greater than or comparable to that of pristine PLA (16.16 J/m). The improved impact performance of the GVC containing PLA composites may be attributed to GVC’s porous structure, which tolerates higher deformation under impact than a solid structure [36]. The results also revealed that larger-sized biochar particles (120 mesh GVC) had a better reinforcing effect on the impact strength of PLA than that of smaller-sized particles (200 mesh). Behazin et al. also reported the same findings. Adding different sizes of grass biocarbon to polypropylene and exploring the effect on the mechanical properties of polypropylene, they discovered that the composite obtained by adding a larger size of 106~125 µm (120 mesh) of biochar has a higher impact strength than a smaller size of 20 µm of biochar. The results of this study are consistent with these findings. Particularly, the smaller the size of the reinforcing material, the higher the likelihood that a better reinforcing effect may not be obtained [37]. Moreover, due to the stiffness-reinforcing effect (increased elastic modulus) of GVC, the heat deflection temperature (HDT) of PLA was increased from 58.6 to 63.2~66.7 °C with the addition of GVC. However, the increase for PLA/120GVC-10 was less significant, indicating that the uneven dispersion of GVC in the matrix led to a lower reinforcing effect.

### 3.5. SEM Analysis of PLA/GVC Composites

Figure 5 illustrates the morphology and detailed microstructure of the PLA/GVC composites, as detected by SEM. The morphology of PLA/200GVC-1 and PLA/120GVC-1 displayed a homogeneous structure, and the good dispersion of GVC can provide an excellent reinforcement effect. However, uneven dispersion and GVC fall off in PLA/200GVC-10 and PLA/120GVC-10 were discovered, particularly in PLA/120GVC-10, which resulted in poor tensile strength.

### 3.6. Molecular Weight Analysis

To analyze the effect of adding GVC on the molecular weight of PLA, GPC measurements were conducted, and the results are shown in Figure 6 and Table 3. The number average molecular weight (M_n_, g/mol) and weight average molecular weight (M_w_, g/mol) of pure PLA were 112,769 and 238,248, respectively. As shown in Table 3, adding a small amount of GVC (1%) did not significantly affect the molecular weight of PLA. M_n_s and M_w_s values slightly decreased to 109,836, 231,084, and 110,350, 227,515 for PLA/120GVC-1 and PLA/120GVC-1, respectively. However, an increase in GVC content to 10% resulted in lower Mns and Mws for both the PLA/GVC200-10 and PLA/GVC120-10 composition. Mns were dropped to 100,740 and 104,056, while Mws were dropped to 189,402 and 199,736 for PLA/GVC 200-10 and PLA/GVC120-10, respectively. Based on our observation, when the content of GVC exceeded 20%, the composite reached a liquefied and non-plastic state, indicating that the molecular chain of PLA had been severely decomposed. In addition, the polydispersity index (PDI) of PLA gradually decreased as the amount of GVC increased. The PDI of pristine PLA was 2.1127, but it decreased to 1.8801 and 1.9195 when the content of GVC was 10%. This is because the addition of GVC may cause chain scission of larger molecules, resulting in a narrower molecular weight distribution of PLA.

## 4. Conclusions

This study found that when GVC is added to PLA, it could promote the crystallization of PLA. The re-crystallization point obviously shifted to a lower temperature as the fineness of the powder and the added amount increased. These GVC/PLA composites exhibit excellent mechanical properties, with tensile and impact strengths of 79.79 MPa and 22.67 J/m, respectively. Moreover, the molecular weight of PLA can be adjusted by the amount of GVC added into PLA. Therefore, they can be recycled or composted with plant residues and be used as agricultural accessories for growing grapes or other plants. Furthermore, these agro-waste-reinforced biodegradable composites can effectively use agricultural waste while achieving the objectives of net zero emissions and the circular economy.

## Figures and Tables

**Figure 1 materials-16-00816-f001:**
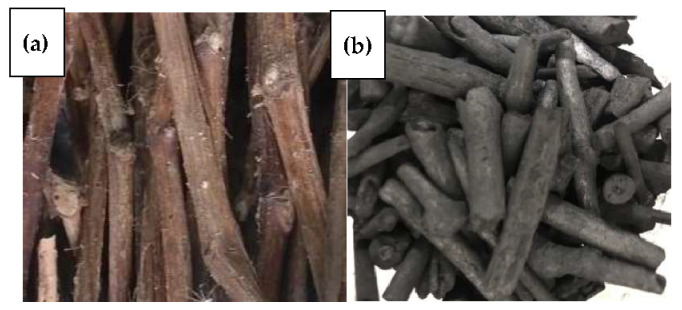
Photographs of (**a**) grapevine, (**b**) grapevine char (GVC).

**Figure 2 materials-16-00816-f002:**
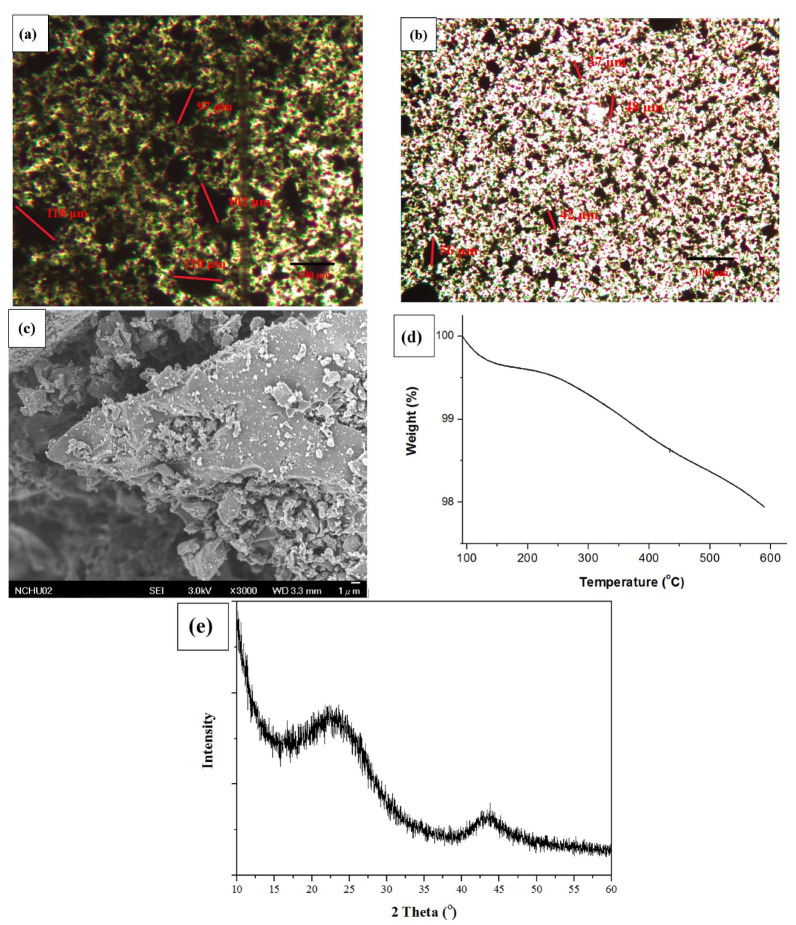
(**a**) POM photograph of 120GVC; (**b**) POM photograph of 200GVC; (**c**) SEM analysis of GVC; (**d**) TGA thermogram of GVC; (**e**) XRD patterns of GVC.

**Figure 3 materials-16-00816-f003:**
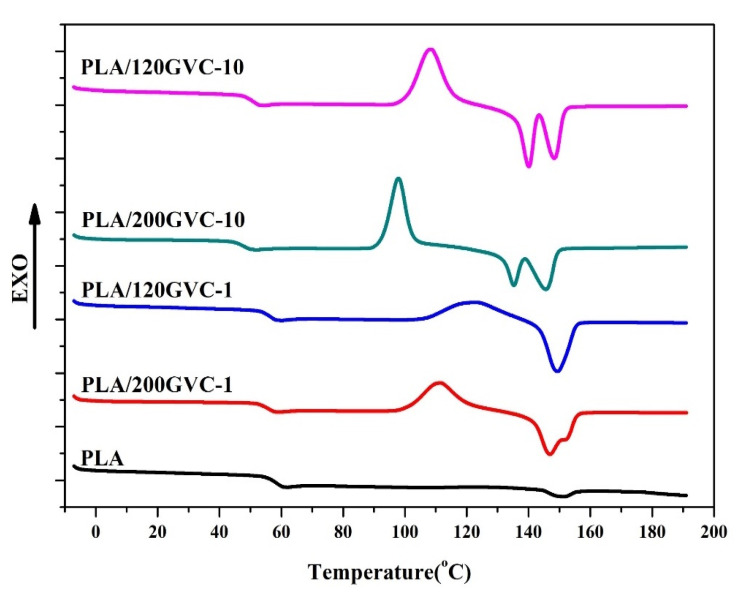
DSC thermograms of PLA and its composites.

**Figure 4 materials-16-00816-f004:**
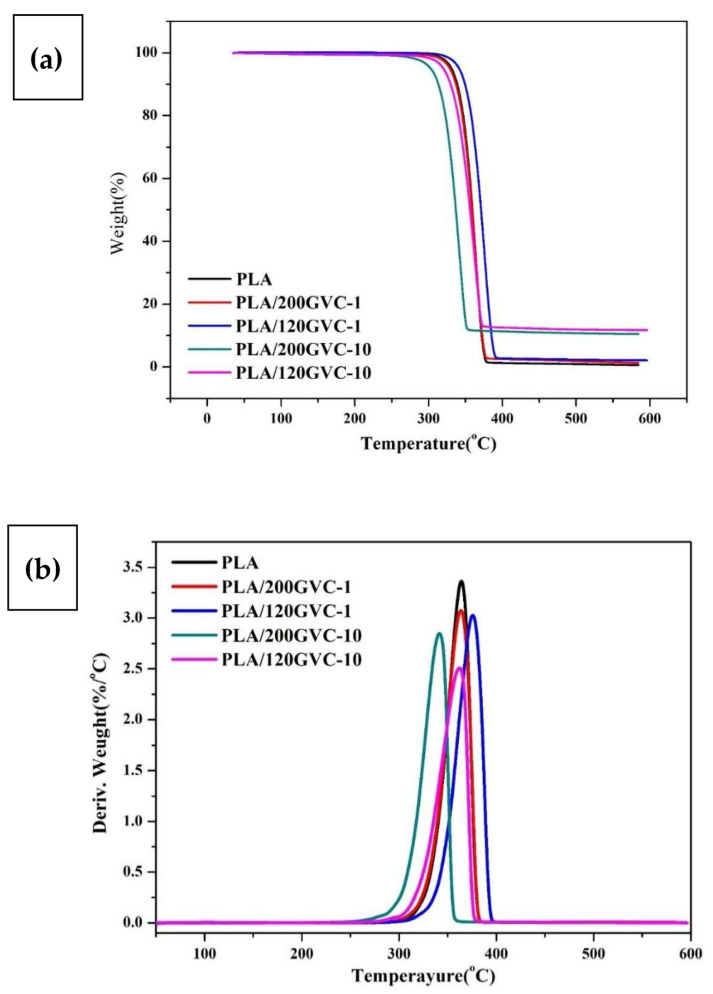
(**a**) TGA and (**b**) DTG thermograms of PLA and its composites.

**Figure 5 materials-16-00816-f005:**
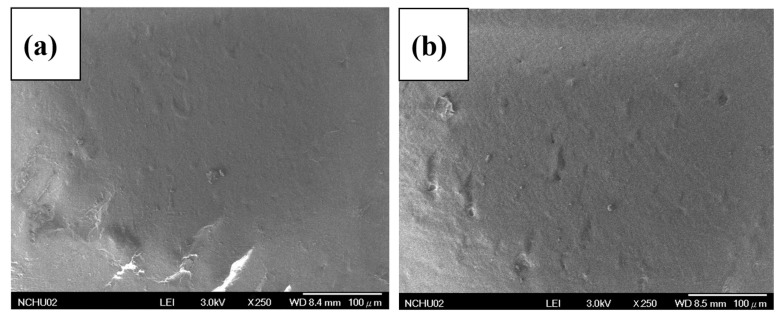
Impact fracture of the (**a**) PLA/200GVC-1; (**b**) PLA/120GVC-1; (**c**) PLA/200GVC-10; (**d**) PLA/120GVC-10.

**Figure 6 materials-16-00816-f006:**
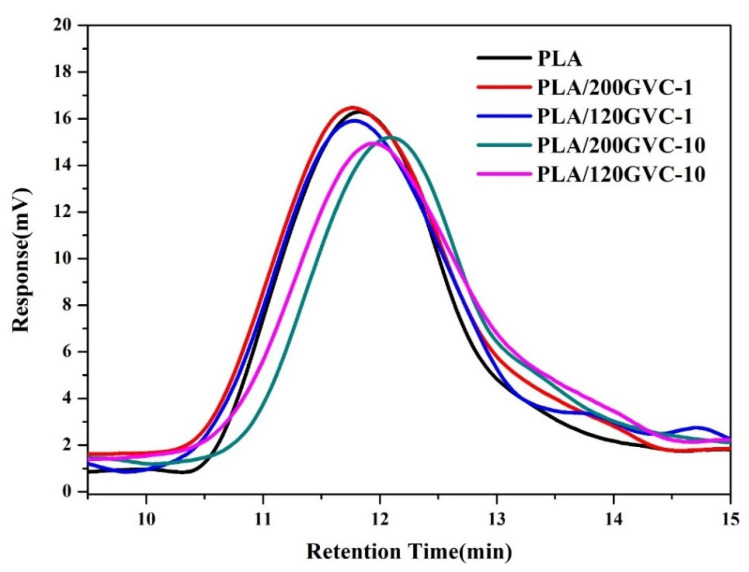
Gel permeation chromatography (GPC) curves of PLA and its composites.

**Table 1 materials-16-00816-t001:** Thermal Properties of PLA and its composites.

Sample	T_g_ (°C)	T_cc_ (°C)	ΔH_cc_ (Jg^−1^)	T_m1_/T_m2_ (°C)	ΔH_m1_/ΔH_m2_ (Jg^−1^)	Crystallinity * (%)	T_d_ (°C)	Char Yield (%)
PLA	60.86	-	-	−/151.09	−/2.456	2.62	364.26	0.62
PLA/200GVC-1	58.03	110.83	15.75	147.02/150.82	35.96/1.82	40.39	364.05	1.33
PLA/120GVC-1	57.78	122.11	16.35	−/149.34	−/20.51	22.09	375.84	1.98
PLA/200GVC-10	53.42	97.85	22.09	135.10/145.66	18.40/23.61	49.76	342.34	10.64
PLA/120GVC-10	53.42	108.28	30.85	140.12/148.62	26.22/22.27	57.44	362.21	11.77

* Crystallinity (%) = [ΔH_m_/ƒ × ΔH_m0_] × 100%. ΔH_m_: enthalpy of melting; ƒ: content of PLA in the composite; ΔH_m0_: standard enthalpy of melting for PLA (93.8 J/g).

**Table 2 materials-16-00816-t002:** Mechanical and Thermal Properties of PLA and Its Composites.

Sample	Tensile Strength (MPa)	Elastic Modulus (GPa)	Impact Strength (J/m)	HDT (°C)
PLA	56.44 ± 0.45	3.725 ± 0.416	16.16 ± 2.10	58.6 ± 1.3
PLA/200GVC-1	65.98 ± 1.46	5.063 ± 0.282	20.00 ± 0.15	65.9 ± 0.3
PLA/120GVC-1	79.79 ± 6.08	4.802 ± 0.240	22.67 ± 2.42	66.7 ± 0.3
PLA/200GVC-10	41.72 ± 2.22	5.268 ± 0.259	15.33 ± 0.71	66.2 ± 0.5
PLA/120GVC-10	29.30 ± 6.64	5.422 ± 0.893	17.11 ± 1.16	63.2 ± 1.4

**Table 3 materials-16-00816-t003:** Molecular Weight Analysis of PLA and Its Composites.

Sample	Mn (g/mol)	Mw (g/mol)	PDI
PLA	112,769	238,248	2.1127
PLA/GVC200-1	109,836	231,084	2.1039
PLA/GVC120-1	110,350	227,515	2.0618
PLA/GVC200-10	100,740	189,402	1.8801
PLA/GVC120-10	104,056	199,736	1.9195

## Data Availability

Not applicable.

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
