# Peer review of "Effects of the Grapevine Biochar on the Properties of PLA Composites"

_materials, 2023, doi:10.3390/ma16020816_

Round 1
Reviewer 1 Report
Dear,
The authors produced vine-based biochar for application as a PLA filler. Therefore, it is about the production of ecological composites, as well as reintroducing biochar in the production chain. The manuscript has merit for publication, but needs adjustments before publication:
> Materials. Please inform the density and melt flow index of the PLA;
> The authors need to add the detailed characterization of the biochar: particle size distribution; thermogravimetry (TG); X-Ray Diffraction (XRD) and SEM;
> Thermal analysis. Inform the mass used and the gas flow;
> Mechanical properties analysis. PLA and composites are fragile. Why did the authors use a speed of 50 mmm/min? Normal would be 5 mm/min. Inform the value of the pendulum used in the impact test;
> DSC analysis. Authors must report the degree of crystallinity of PLA and composites;
> TG. Please discuss more clearly how biochar accelerates the thermal degradation of PLA. Did biochar act as a catalyst accelerating degradation?
> Table 2. Authors need to add elastic modulus results in order to understand stiffness;
> PLA/200GVC-1 and PLA/120GVC-1 composites showed excellent reinforcement effect. Authors need to add Scanning Electron Microscopy (SEM) of biochar and composites. This will help you understand the reinforcement effect;
Author Response
Response to the comments of reviewer 1 (#materials-2097294)
We appreciate the referee’s careful reading and thoughtful suggestions. Point-by-point responses to those comments are attached along with highlighted changes illustrated in the text accordingly.
- Materials. Please inform the density and melt flow index of the PLA ;
Authors’ reply:
We are grateful for the reviewer’s suggestion. The density and melt flow index of the PLA have been newly added as follows:
NatureWorks, LLC provided the polylactic acid (PLA, PLA2003D) with a density of 1.24 g/cm3, melt flow index of 6.0 g/10 min at 210°C, and a load of 2.16 kg.
- The authors need to add the detailed characterization of the biochar: particle size distribution; thermogravimetry (TG); X-Ray Diffraction (XRD) and SEM;
Authors’ reply:
We are grateful for the reviewer’s suggestion. The particle size distribution; thermogravimetry (TG); X-Ray Diffraction (XRD) and SEM of biochar has been newly added as follows:
Figure 2 (a)~(e) shows the particle size distribution, SEM, TGA, and X-Ray Diffraction (XRD) analysis of the GVC biochar. The particle sizes of 120GVC and 200GVC were 97~118 and 37~56 µm, respectively, as shown in the POM analysis in Figure 2 (a) and (b). The morphology of GVC (Figure 2(c)) exhibited a porous structure. TGA analysis (Figure 2 (d)) showed that the char yield was high as 98%, indicating a low organic content. On the XRD patterns of GVC (Figure 2(e)), the graphitic peaks were found around the 2Ï´ values of 25°(002) and 44°(10) [33]. However, the low intensity and inconspicuous peak revealed that the GVC biochar was an amorphous structure.
Figure 2. (a)POM photograph of 120GVC (b)POM photograph of 200GVC (c)SEM analysis of GVC (d)TGA thermogram of GVC (e)XRD patterns of GVC
- Li , Z.Q.; Lu, C.J.; Xia, Z.P. ; Zhou, Y.; Luo, Z. X-ray diffraction patterns of graphite and turbostratic carbon, Carbon 2007, 45, 1686–1695, doi:10.1016/j.carbon.2007.03.038.
- Thermal analysis. Inform the mass used and the gas flow;
Authors’ reply:
We are grateful for the reviewer’s suggestion. The mass used and the gas flow have been newly added as follows:
Samples of 5-10 mg were heated from room temperature to 190℃ at a heating rate of 10 ℃/min and a nitrogen flow rate of 50 cm3/min.
- Mechanical properties analysis. PLA and composites are fragile. Why did the authors use a speed of 50 mmm/min? Normal would be 5 mm/min. Inform the value of the pendulum used in the impact test;
Authors’ reply:
We are grateful for the reviewer’s suggestion.
(1) “50 mm/min” has been corrected to “5 mm/min”.
(2) The impact strength of the composite material was studied using an impact resistance testing machine (GT-70045-MDL, Gotech Testing Machines Co., Taiwan), with an initial potential energy of 2.7 J at a specified drop height of 610 ±2 mm, following ASTM D256.
- DSC analysis. Authors must report the degree of crystallinity of PLA and composites;
Authors’ reply:
The degree of crystallinity of PLA and composites has been newly added as follows:
Furthermore, the crystallinity analysis revealed that PLA crystallinity increased with the GVC content (from 2.62% to 40.39, 22.09, 49.76, and 57.44%), indicating that GVC serves as an important nucleating agent.
|
Sample |
Tg (℃) |
Tcc (℃) |
ΔH cc (Jg-1) |
Tm1 /Tm2 (℃) |
ΔH m1 / ΔH m2 (Jg-1) |
Crystallinity* (%) |
Td (℃) |
Char yield (%) |
|
PLA |
60.86 |
- |
- |
- /151.09 |
-/2.456 |
2.62 |
364.26 |
0.62 |
|
PLA/200GVC-1 |
58.03 |
110.83 |
15.75 |
147.02/150.82 |
35.96/1.82 |
40.39 |
364.05 |
1.33 |
|
PLA/120GVC-1 |
57.78 |
122.11 |
16.35 |
-/149.34 |
-/20.51 |
22.09 |
375.84 |
1.98 |
|
PLA/200GVC-10 |
53.42 |
97.85 |
22.09 |
135.10/145.66 |
18.40/23.61 |
49.76 |
342.34 |
10.64 |
|
PLA/120GVC-10 |
53.42 |
108.28 |
30.85 |
140.12/148.62 |
26.22/22.27 |
57.44 |
362.21 |
11.77 |
*Crystallinity (%) = [ΔHm / ƒΔHm0 ]100%
ΔHm: enthalpy of meltingï¼›ƒ: content of PLA in the compositeï¼›ΔHm0: standard enthalpy of melting for PLA (93.8 J/g)
- TG. Please discuss more clearly how biochar accelerates the thermal degradation of PLA. Did biochar act as a catalyst accelerating degradation?
Authors’ reply:
We are grateful for the reviewer’s suggestion. The following discuss has been newly added to the text:
The molecular weight analysis confirms that the PLA chain was decomposed by the GVC catalyzation, resulting in decreased decomposition temperature.
- Table 2. Authors need to add elastic modulus results in order to understand stiffness;
Authors’ reply:
The elastic modulus of PLA and its composites has been newly added to Table 2 as follows:
In addition, the elastic modulus of PLA was increased from 3.725 to 4.802~5.422 GPa with the addition of GVC. This revealed that the addition of GVC improved the stiffness of PLA.
Table 2 Mechanical and thermal properties of PLA and its composites
|
Sample |
Tensile strength (MPa) |
Elastic modulus (GPa) |
Impact strength (J/m) |
HDT (℃) |
|
PLA |
56.44±0.45 |
3.725±0.416 |
16.16±2.10 |
58.6±1.3 |
|
PLA/200GVC-1 |
65.98±1.46 |
5.063±0.282 |
20.00±0.15 |
65.9±0.3 |
|
PLA/120GVC-1 |
79.79±6.08 |
4.802±0.240 |
22.67±2.42 |
66.7±0.3 |
|
PLA/200GVC-10 |
41.72±2.22 |
5.268±0.259 |
15.33±0.71 |
66.2±0.5 |
|
PLA/120GVC-10 |
29.30±6.64 |
5.422±0.893 |
17.11±1.16 |
63.2±1.4 |
- PLA/200GVC-1 and PLA/120GVC-1 composites showed excellent reinforcement effect. Authors need to add Scanning Electron Microscopy (SEM) of biochar and composites. This will help you understand the reinforcement effect;
Authors’ reply:
We are grateful for the reviewer’s suggestion. The SEM analysis has been newly added to the text:
Figure 6 illustrates the morphology and detailed microstructure of the PLA/GVC composites, as detected by SEM. The morphology of PLA/200GVC-1 and PLA/120GVC-1 displayed a homogeneous structure, and the good dispersion of GVC can provide an excellent reinforcement effect. However, uneven dispersion and fall off of GVC in PLA/200GVC-10 and PLA/120GVC-10 were discovered, particularly in PLA/120GVC-10, which resulted in poor tensile strength.
Figure 6. Impact fracture of the (a)PLA/200GVC-1 (b)PLA/120GVC-1 (c)PLA/200GVC-10 (d)PLA/120GVC-10

Reviewer 2 Report
Study is interesting and can be published after major modifications.
Add Morphological analysis of the composites.
The particle size analysis of the biochar is missing.
The thermal analysis must be added to analyse the effect of biochar reinforcement.
Grammatical mistakes were observed, manuscript must be thoroughly revised.
Author Response
Response to the comments of reviewer 2 (#materials-2097294)
We appreciate the referee’s careful reading and thoughtful suggestions. Point-by-point responses to those comments are attached along with highlighted changes illustrated in the text accordingly.
- Add Morphological analysis of the composites.
Authors’ reply:
We are grateful for the reviewer’s suggestion. The SEM analysis has been newly added to the text:
Figure 6 illustrates the morphology and detailed microstructure of the PLA/GVC composites, as detected by SEM. The morphology of PLA/200GVC-1 and PLA/120GVC-1 displayed a homogeneous structure, and the good dispersion of GVC can provide an excellent reinforcement effect. However, uneven dispersion and fall off of GVC in PLA/200GVC-10 and PLA/120GVC-10 were discovered, particularly in PLA/120GVC-10, which resulted in poor tensile strength.
Figure 6. Impact fracture of the (a)PLA/200GVC-1 (b)PLA/120GVC-1 (c)PLA/200GVC-10 (d)PLA/120GVC-10
- The particle size analysis of the biochar is missing.
Authors’ reply:
We are grateful for the reviewer’s suggestion. The particle size distribution; thermogravimetry (TG); X-Ray Diffraction (XRD) and SEM of biochar has been newly added as follows:
Figure 2 (a)~(e) shows the particle size distribution, SEM, TGA, and X-Ray Diffraction (XRD) analysis of the GVC biochar. The particle sizes of 120GVC and 200GVC were 97~118 and 37~56 µm, respectively, as shown in the POM analysis in Figure 2 (a) and (b). The morphology of GVC (Figure 2(c)) exhibited a porous structure. TGA analysis (Figure 2 (d)) showed that the char yield was high as 98%, indicating a low organic content. On the XRD patterns of GVC (Figure 2(e)), the graphitic peaks were found around the 2Ï´ values of 25°(002) and 44°(10) [33]. However, the low intensity and inconspicuous peak revealed that the GVC biochar was an amorphous structure.
Figure 2. (a)POM photograph of 120GVC (b)POM photograph of 200GVC (c)SEM analysis of GVC (d)TGA thermogram of GVC (e)XRD patterns of GVC
- Li , Z.Q.; Lu, C.J.; Xia, Z.P. ; Zhou, Y.; Luo, Z. X-ray diffraction patterns of graphite and turbostratic carbon, Carbon 2007, 45, 1686–1695, doi:10.1016/j.carbon.2007.03.038.
- The thermal analysis must be added to analyse the effect of biochar reinforcement.
Authors’ reply:
The heat deflection temperature (HDT) of PLA and its composites has been newly added to Table 2 as follows:
Moreover, due to the stiffness-reinforcing effect (increased elastic modulus) of GVC, the heat deflection temperature (HDT) of PLA was increased from 58.6 to 63.2~66.7℃ with the addition of GVC. However, the increase for PLA/120GVC-10 was less significant, indicating that the uneven dispersion of GVC in the matrix led to a lower reinforcing effect.
Table 2 Mechanical and thermal properties of PLA and its composites
|
Sample |
Tensile strength (MPa) |
Elastic modulus (GPa) |
Impact strength (J/m) |
HDT (℃) |
|
PLA |
56.44±0.45 |
3.725±0.416 |
16.16±2.10 |
58.6±1.3 |
|
PLA/200GVC-1 |
65.98±1.46 |
5.063±0.282 |
20.00±0.15 |
65.9±0.3 |
|
PLA/120GVC-1 |
79.79±6.08 |
4.802±0.240 |
22.67±2.42 |
66.7±0.3 |
|
PLA/200GVC-10 |
41.72±2.22 |
5.268±0.259 |
15.33±0.71 |
66.2±0.5 |
|
PLA/120GVC-10 |
29.30±6.64 |
5.422±0.893 |
17.11±1.16 |
63.2±1.4 |
- Grammatical mistakes were observed, manuscript must be thoroughly revised.
Authors’ reply:
The manuscript has been proofread by Charles Maratas, who is a native English speaker, prior to submission. In addition, the title of this manuscript has been revised to “Effects of the grapevine biochar on the properties of PLA composites”, and the proof is as follows:

Round 2
Reviewer 1 Report
The authors improved the quality of the manuscript, generating greater clarity. Therefore, the manuscript has merit for publication.
Yours sincerely,
Author Response
Response to the comments of reviewer 1 (#materials-2097294 Round 2)
We appreciate the referee’s careful reading and thoughtful suggestions.
- The authors improved the quality of the manuscript, generating greater clarity. Therefore, the manuscript has merit for publication.
Authors’ reply:
We are grateful for the reviewer’s affirmation.
2. The manuscript has been proofread by Charles Maratas, who is a native English speaker, prior to submission.

Reviewer 2 Report
Figure 5 ans Table 2 contents the same results. I suggest the removal of Figure 5 because it contains HDT values, which confuses the readers, as double y axis have duel naming.
Author Response
Response to the comments of reviewer 2 (#materials-2097294 Round 2)
We appreciate the referee’s careful reading and thoughtful suggestions. Point-by-point responses to those comments are attached along with highlighted changes illustrated in the text accordingly.
- Figure 5 and Table 2 contents the same results. I suggest the removal of Figure 5 because it contains HDT values, which confuses the readers, as double y axis have duel naming.
Authors’ reply:
We are grateful for the reviewer’s suggestion. Figure 5 has been removed.